# Regenerative Potential of Carbon Monoxide in Adult Neural Circuits of the Central Nervous System

**DOI:** 10.3390/ijms21072273

**Published:** 2020-03-25

**Authors:** Eunyoung Jung, Seong-Ho Koh, Myeongjong Yoo, Yoon Kyung Choi

**Affiliations:** 1Department of Bioscience and Biotechnology, Konkuk University, Seoul 05029, Korea; eunyoungjjjjj@gmail.com (E.J.); rkcldk4@naver.com (M.Y.); 2Department of Neurology, Hanyang University Guri Hospital, Guri 11923, Korea; ksh213@hanyang.ac.kr

**Keywords:** carbon monoxide, neurogenesis, angiogenesis, regeneration, stroke, traumatic brain injury, multiple sclerosis, Alzheimer’s disease, central nervous system

## Abstract

Regeneration of adult neural circuits after an injury is limited in the central nervous system (CNS). Heme oxygenase (HO) is an enzyme that produces HO metabolites, such as carbon monoxide (CO), biliverdin and iron by heme degradation. CO may act as a biological signal transduction effector in CNS regeneration by stimulating neuronal intrinsic and extrinsic mechanisms as well as mitochondrial biogenesis. CO may give directions by which the injured neurovascular system switches into regeneration mode by stimulating endogenous neural stem cells and endothelial cells to produce neurons and vessels capable of replacing injured neurons and vessels in the CNS. The present review discusses the regenerative potential of CO in acute and chronic neuroinflammatory diseases of the CNS, such as stroke, traumatic brain injury, multiple sclerosis and Alzheimer’s disease and the role of signaling pathways and neurotrophic factors. CO-mediated facilitation of cellular communications may boost regeneration, consequently forming functional adult neural circuits in CNS injury.

## 1. Overview of Repairing the Damaged Brain and Heme Oxygenase (HO) Metabolites

Some devastating brain disorders are the result of disturbances in the formation of neural circuits in embryonic or postnatal development. In the peripheral nervous system and the central nervous system (CNS) of lower vertebrates, there is robust regeneration of severed axons [1,2,3]. However, regeneration following CNS injury in higher mammals is very poor [4]. Investigators have tried to discover precise mechanisms to augment the limited ability of CNS neurons to recover normal functions [5,6]. The purpose of this work was to decipher the crucial barriers to regeneration, so that they may be overcome [7]. One critical question regarding the poor regeneration of central axons is whether the low recovery reflects an inability of neurons themselves to grow or an inability of the environment to support axonal growth [5]. Therefore, in this review, strategies for overcoming CNS injury through the evaluation of (1) the intrinsic ability of neurons, (2) environmental support via vascular and glial cells and (3) regenerative pathways, including the vascular system, neural stem cell (NSC)-mediated neurogenesis and mitochondrial biogenesis, will be introduced.

One of our focused therapeutic strategies is heme oxygenase (HO) metabolite-mediated CNS regeneration [8,9]. HO metabolites are biliverdin, carbon monoxide (CO) and ferrous iron (Fe^2+^), which are byproducts of heme catabolism by HO [10,11]. Biliverdin is a tetrapyrrolic bile pigment that is further reduced to the antioxidant bilirubin by the enzyme biliverdin reductase using nicotinamide adenine dinucleotide (NADH) and NADH phosphate (NADPH) as electron donors [12,13]. Iron triggers pro-oxidant effects and cells respond to its release by upregulating the expression of the iron storage protein, ferritin [14]. Accumulation of excess iron in mitochondria as a consequence of astrocyte-specific long-term overexpression of HO-1 results in cellular toxicity [15]. HO-1 is an isoform and inducible form of HO and HO-2 is a constitutive form [16]. HO-3 may be a pseudogene derived from HO-2 transcripts in rats and is non-functional [17].

HO-1 can be induced by oxidative stress, low oxygen tension and gaseous molecules, such as nitric oxide (NO) and CO [10]. The relationship between HO and adult neural circuit regeneration seems to be complex [18]. Long-term overexpression of HO-1 may be detrimental for cognitive ability in the rodent Alzheimer’s disease (AD) model [15,19]. Abnormal patterns of iron deposition caused by long-term glial-HO-1 overexpression can cause cytotoxicity in astrocytes via mitochondrial dysfunction, leading to neuronal toxicity [15]. In contrast, transient HO-1 activation via CO treatment may be beneficial for regeneration in acute ischemic CNS injury [8,9,18,20,21,22].

CO is an endogenous biological molecule that exerts beneficial effects by upregulating HO-1 expression [10]. CO has been demonstrated to act as a neurophysiological signaling molecule without neurotoxicity in healthy men [23]. CO can also be considered as a neurotransmitter [24], a vascular tone modulator [25] and an anti-inflammatory factor [8]. The CO-HO-1 circuit may further produce HO metabolites, which can be found in nervous tissue [26]. To achieve transient increases in HO-1 expression, CO-releasing molecules (CORMs) have been used in various injury models. Therefore, we will introduce the effects of CORMs on acute and chronic CNS diseases, such as stroke, traumatic brain injury (TBI), multiple sclerosis and AD.

## 2. Biological Signaling of CO: Use of CO Gas and CORMs

Among the diverse HO metabolites (e.g., CO, biliverdin, bilirubin and iron), we will focus on the role of CO in regenerative potential. Physiological concentrations of CO derived from basal production of HO have been shown to facilitate signaling processes related to the brain [27]. Using CORMs, it has been possible to demonstrate that controlled amounts of CO at low micromolar concentrations delivered to isolated mitochondria, cells or tissues unexpectedly increase mitochondrial functions (i.e., enhanced O_2_ consumption) and that this effect may play a key role in important redox, metabolic and regenerative processes [28,29,30,31]. Application of CORMs recapitulates many of the cytoprotective effects observed following HO-1 induction [10], which results in enhanced production of HO metabolites such as CO and bilirubin. Among CORMs, diversified chemical structures lead to various amounts of CO release to biological systems [32]. CORM-1 and CORM-2 are lipid-soluble CORMs [8] (Figure 1). CORM-A1, CORM-3, CORM-401, methyldiphenylsilacarboxylic acid and CORM ALF-186 are water-soluble CORMs [22,33] (Figure 1). Application of the same concentrations of structurally different CORMs does not elicit the same fluorescent signal of CO [28]. Intracellular delivery of CO by CORM-3 and CORM-401 is significantly enhanced but that of CO by CORM-A1 is negligible in endothelial cells [27]. This result is likely due to the distinct rates of CO release by these compounds, as well as the specific chemical reactivity of each CORM with cellular components [27].

CORMs have protective and anti-inflammatory effects in CNS injury animal models [34,35,36]. In addition, CORMs may induce regeneration through stimulating angiogenesis or/and neurogenesis following CNS injury [9,22,37,38]. The CO-HO-1 axis increases angiogenesis by upregulating angiogenic stimulators such as interleukin (IL)-8, vascular endothelial growth factor (VEGF) and stromal cell-derived factor-1 [10]. Moreover, the CO-HO-1 pathway increases ATP production through mitochondrial biogenesis [31,39,40,41], possibly establishing environmental conditions for axonal growth [42].

In this review, we suggest that the CO-HO-1 pathway may affect (1) the intrinsic pathways of neuronal axon regeneration, (2) the expression of neurotrophic factors (i.e., brain-derived neurotrophic factor (BDNF), glial cell-line derived neurotrophic factor (GDNF) and ciliary neurotrophic factor (CNTF)) and (3) axonal regeneration through adult neurogenesis, angiogenesis and mitochondrial biogenesis after CNS injuries. We hypothesized that CO may have beneficial effects through the induction of transient HO-1 expression, crosstalk between CO-HO and NO-NO synthase (NOS) and consequent induction of regenerative signaling pathways. In addition, CO bound with ferrous hemoproteins (i.e., guanylyl cyclase and NADPH oxidase (NOX)) serves as a posttranslational modification that regulates important processes, including aerobic metabolism, oxidative stress and mitochondrial bioenergetics in the CNS [43]. In this review, we highlight the cellular and molecular mechanisms of the CO-HO pathway that form adult neural circuits in the CNS.

## 3. Effects of CO on Neuronal Intrinsic Mechanisms

The ability of a CNS neuron to regenerate following injury may be dependent on both its intrinsic growth capacity and the extracellular environment enriched by the cell-cell network [5,6,7,44]. In this section, the intrinsic mechanisms of neuronal axon regeneration will be discussed. The role of the CO-HO pathway in this intrinsic pathway has not been well reported; however, some reports may give clues to the association of CO with the neural intrinsic pathway [43].

In injured neurons, bursting of the axonal membrane potential combined with the opening of voltage-dependent Na^+^ channels and the inversion of Na^+^/Ca^2+^ exchange pumps promotes intracellular calcium waves at the injury site through axon-soma communication [45]. A Ca^2+^ wave is propagated to the soma through L-type Ca^2+^ channels and the release of Ca^2+^ from the endoplasmic reticulum via ryanodine receptors, consequently inducing epigenetic changes [46,47]. Catastrophic death signaling cascades can be initiated by excessive concentrations of Ca^2+^; however, moderate levels of intracellular Ca^2+^ can play key roles in intrinsic axon regeneration by inducing membrane resealing, growth cone assembly and maturation, translocation and activation of specific epigenetic modifiers and protein synthesis and activation of signaling molecules and transcription factors [47,48,49,50].

One multifaceted signaling molecule downstream of Ca^2+^ is the second messenger cyclic adenosine monophosphate (cAMP) [51]. Generated by Ca^2+^-dependent enzyme adenylyl cyclase, cAMP initiates the activity of several important pro-regenerative proteins, including dual leucine zipper-bearing kinase (DLK, also known as MAP3K12), an important pro-regenerative kinase [52]. Protein kinase A (PKA), activated by the Ca^2+^-cAMP pathway, can activate DLK, which is critical for axon-to-soma retrograde signaling and axon regeneration following a nerve injury [53]. The sequential Ca^2+^-cAMP-PKA axis activates a cAMP-responsive element-binding protein (CREB), an important axon regeneration- and synaptogenesis-promoting transcription factor [49,54]. Recently, increased CREB-regulated transcription coactivators, such as transcription factor activator protein-1 (c-Jun), have emerged as key regulators for the regeneration of central axons following injury [55,56,57]. In addition, neuronal activity-induced molecules involve the Ca^2+^-dependent nuclear export of histone deacetylase 5 (HDAC5) and the activation of CREB-binding protein and mitogen-activated protein kinases (MAPKs) [6]. The backpropagation of Ca^2+^ to the cell soma stimulates the translocation of the serine/threonine-protein kinase D1 to the nucleus, which triggers HDAC5 nuclear export [47]. The translocation of HDAC5 to the cytosol in injured neurons promotes axon regeneration [47,58]. Phosphorylation of HDAC5 by AMP kinase (AMPK) promotes the shuttling of HDAC5 from the nucleus to the cytosol, in which cytosolic HDAC5 activity facilitates hypoxia-inducible factor-1α (HIF-1α) stabilization via the heat shock protein 90 (Hsp90) and rapid nuclear accumulation [59,60]. CORM-2 facilitates the binding of Hsp90 with HIF-1α, consequently stabilizing HIF-1α protein by inhibiting proteasome-mediated HIF-1α degradation in astrocytes [37]. Neuronal Hsp90 can bind with various client proteins and enhance axonal growth and functional recovery in CNS injured mice [61]. Therefore, to demonstrate the role of CO in the intrinsic mechanism of axon regeneration, it is necessary to conduct experiments showing the effects of CO on HDAC5 activation and binding between Hsp90 and HIF-1α in neurons.

Ca^2+^ at the injury site facilitates the local translation of the axonal signal transducer and activator of transcription 3 (STAT3) [62]. Injury can trigger the phosphorylation of STAT3, possibly through the activity of cytokines and phosphorylated STAT3 (p-STAT3) can localize either to the mitochondria as an S727 p-STAT3 to increase ATP synthesis or to the nucleus as a Y705 p-STAT3 to influence transcription of pro-regenerative genes [63]. Both Janus kinases (JAKs) and MAPK kinases (MEKs) phosphorylate STAT3: JAKs phosphorylate tyrosine 705 residue and MEKs phosphorylates serine 727 residue [63]. In hyperoxic endothelial cells, the protective effects of HO-1 and exogenous CO are partly dependent on STAT3 and STAT3 can also activate HO-1 to form a positive feedback loop [64]. In neurons, however, the STAT3-HO-1 circuit has not been well studied.

In the intrinsic pathway, growth-associated protein 43 (GAP43) plays key roles in axon growth and neuronal plasticity in the CNS, including retinal ganglion cells (RGCs) [22,65]. GAP43 synthesis is increased after CNS nerve injury, potentially recapitulating development [66,67]. During the regeneration period, GAP43 expression in the perihematomal tissues is also upregulated when rats are surged with bone marrow mesenchymal stem cell transplantation at 1 and 24 h after intracerebral hemorrhage [68], contributing to axon regeneration. A recent report suggests that CO enhances GAP43 expression during the regeneration period. Intravitreal injection of CORM ALF-186 significantly upregulated GAP43 mRNA expression and RGC regeneration in rat ischemic/reperfusion vascular injury [22]. Moreover, ALF-186 also exerts neuroprotective effects by reducing the expression of inflammatory mediators, such as nuclear factor kappa-light-chain-enhancer of activated B cells (NF-κB), tumor necrosis factor-α (TNF-α) and IL-6 via the soluble guanylate cyclase β1 in rat RGCs after ischemic/reperfusion vascular injury [69]. In this ischemic model, ALF-186 upregulates Hsp90 mRNA and protein levels [69], possibly affecting the intrinsic mechanism of axon regeneration.

Collectively, the role of CO in axon regeneration via the intrinsic pathway has not been well established; however, recent data using CORM-injected animal experiments suggest that CO may regulate various molecules related to the intrinsic pathway. Therefore, the therapeutic capacity of CO in the overall intrinsic pathway needs further investigation.

## 4. Neurovascular Regeneration

Some vessels and neurons in the adult brain and retina can be replaced by new ones in pathophysiologic conditions [70,71]. New vessels are generated from existing vessels during the process of angiogenesis [72]. Angiogenesis involves the sprouting, migration, growth and differentiation of endothelial cells [73,74]. Recently, the important role of lymphatic vessels in the drainage system for waste removal in the CNS has emerged [75,76,77]. Healthy lymphatic vessels remove macromolecules such as beta-amyloid (Aβ), consequently maintaining functional CNS neural circuit [76,78]. Aging and CNS injury diminish the clearance ability of lymphatic vessels [78,79,80]. In the normal adult CNS, new neurons are added to the brain for the maintenance of learning and memory. NSCs are proliferated, integrated, migrated and differentiated and then new neurons can be generated [7]. The new neurons extend axons and dendrites, form synapses and become integrated into functional circuits [81]. In this section, we will discuss regenerative signaling related to angiogenesis, lymphangiogenesis, neurogenesis and mitochondrial biogenesis, which may be stimulated by the CO-HO-1 pathway in the CNS neurovascular system.

### 4.1. Vascular Functions

In pathophysiological conditions, VEGF and its receptors are important regulators of lymphangiogenesis, vasculogenesis and angiogenesis, as well as vascular permeability in vertebrates [82,83]. VEGF-A, the prototype VEGF ligand, binds and activates two tyrosine kinase receptors: VEGF receptor 1 (VEGFR1) and VEGFR2. VEGFR2 is the major signal initiator for angiogenesis and specially uses the phospholipase C γ-protein kinase C-MAPK pathway [84]. The two transcription factors for VEGF-A expression are HIF-1α and peroxisome proliferator-activated receptor γ-coactivator-1α (PGC-1α)/estrogen-related receptor α (ERRα) [85,86]. Therefore, angiogenesis can be generated both by an O_2_-dependent manner through the HIF-1α-VEGF axis and an O_2_-independent manner through the PGC-1α-ERRα-VEGF axis. In pathophysiologic CNS, astrocytes are the main VEGF-producing cell type [87,88] and astrocytes-derived VEGF secretion can lead to vasculogenesis and angiogenesis. Direct treatment of astrocytes with CORM-2 can upregulate the synthesis of HIF-1α protein through the phosphatidylinositide 3-kinase (PI3K)-Akt (protein kinase B) pathway and MAPK-extracellular-signal-related kinase (ERK) pathway and the CO/HO-1 circuit activates those pathways [37]. Pretreatment of astrocytes with CORM-2 following recovery (CO/R) stabilizes both HIF-1α and PGC-1α protein by inhibiting proteasome-mediated ubiquitination [38,40]. Direct treatment of endothelial cells with CORM-401 induces endothelial cell migration through p38 MAPK activation and increases angiogenic molecules, such as VEGF, IL-8 and HO-1 [28].

Lymphatic vessels are lined by lymphatic endothelial cells, which are distinct from blood endothelial cells in that they acquire and maintain their identity by a lymphatic master transcription factor, prospero homeobox 1 (Prox1), which is negatively regulated by effectors of the Hippo pathway such as Yes-associated protein (YAP) and transcriptional coactivator with PDZ-binding motif (TAZ) [89,90]. The mammalian Hippo signaling pathway and its downstream effectors YAP/TAZ are important regulators of sprouting angiogenesis, maturation of the blood-brain barrier and lymphangiogenesis [89,91,92]. YAP promotes angiogenesis via STAT3 in endothelial cells [93]. The role of CO in Hippo pathway-mediated lymphangiogenesis has not been demonstrated. Since Prox1 interacts with ERRα and PGC-1α forming a trimeric complex and influences the transcriptional activity of ERRα [94], CO may be involved in Prox1 activation by modulating protein complexes among PGC-1α, Prox1 and ERRα in lymphatic endothelial cells. Thus, the role of CO in lymphatic vessels needs to be investigated.

Healthy vessels, including blood vessels and lymphatic vessels, are quite important for neuroprotection and regeneration [77,95]. Waste products such as Aβ and tau are removed from tissues via blood vessels and lymphatic vessels [77]. A recent report demonstrates that meningeal lymphatic vessels at the skull base act as a drainage system for toxic metabolites [77]. Dysfunction of vascular and lymphatic systems with age can lead to the accumulation of toxic aggregates within the brain, consequently causing neurodegenerative diseases such as AD [78]. Chronic hypoxia or aging revealing a damaged vascular system leads to diminished oxygen supply to surrounding tissues, reduced metabolism and inhibition of drainage of toxic protein aggregates [77,96]. CORM-3 can protect vascular permeability via suppression of inflammatory responses in the stroke [36,97] and TBI models [9]. Therefore, functional vascular repair by CO after ischemic injury may contribute to neuroprotection and regeneration, possibly through angiogenesis and lymphangiogenesis.

### 4.2. Neuroprotection and Neurogenesis

Besides the intrinsic pathway in axons, new neurons can be substituted into injured adult neurons through NSC proliferation and differentiation [9]. Neurovascular communication appears to play a key role in adult neurogenesis, which is generated in the subventricular zone (SVZ) lining the lateral ventricles and the subgranular zone (SGZ) within the dentate gyrus (DG) of the hippocampus [98]. Adult human hippocampal neurogenesis occurs to generate new neurons that play a key role in learning and memory [99]. In pathologic conditions, however, adult hippocampal neurogenesis is impaired [100].

The neuroprotective role of CO in injury in vitro models has been reported [101,102,103]. Treatment of immortalized mouse NSCs with CORM-2 reduces iron overload-mediated ferroptosis via the inhibition of NF-κB activation [101]. Neurons overexpressing HO-1 under neuron-specific enolase control reduces H_2_O_2_-mediated cell death [102]. Application of human neuronal cells with CORM-2 or the HO inducer hemin recovers neurite elongation inhibited by inflammatory conditioned microglia [103]. The neurogenic effects of CO have been demonstrated. Treatment of neuroblastoma cells with CORM-A1 possesses neuroprotective and neurogenic effects [104]. New CORM (MePh_2_SiCO_2_H) or short-term exposure to low doses (25 ppm) of CO gas stimulates differentiation of human NSCs into dopaminergic neurons [33]. CORM-3 inhibits pericyte apoptosis and establishes brain pericytes-NSC crosstalk 3 d after TBI, which may enhance NSC proliferation, migration and neuronal differentiation in SVZ and SGZ at 14 d after TBI [9]. Treatment of pericytes with oxygen-glucose deprivation (OGD) plus CORM-3 promotes NSC proliferation and differentiation into mature neurons in hypoxia/reoxygenation conditions [9]. Hence, CO may induce adult neuroprotection and neurogenesis via direct actions in NSCs and indirect actions through cell-cell interactions.

### 4.3. Mitochondria Function

Neurons traffic mitochondria to regions of high ATP consumption and cytosolic Ca^2+^ concentrations for axonal growth and buffering the excessive intracellular Ca^2+^, respectively [42]. For axon growth, actin must be ATP loaded to establish polymerization and nucleation. Therefore, mitochondria can be transported to the growth cone in the anterograde (soma-to-axon) direction, can commence movement and shape changes within the growth cone, undergo fission and fusion and can also be transported in the retrograde (axon-to-soma) direction [105]. Precise regulation of mitochondrial transport during axon regeneration and axon extension has begun to be elucidated. Overexpression of armadillo repeat-containing X-linked 1 (Armcx1) in adult mice increases mitochondrial transport and enhances adult RGC survival and axon regeneration [106]. Phosphatase and tensin homolog (PTEN) deletion shows an elevated mammalian target of rapamycin (mTOR) activity. Armcx1 overexpression under PI3K activated conditions using PTEN deficient mice significantly increases RGC survival and axon regeneration, which depend on mitochondrial localization [106]. Impaired mitochondrial functions by OGD can be restored through the activation of the PI3K-Akt pathway and the inhibition of excessive Ca^2+^ influx in NSCs [107].

Healthy mitochondria have a highly hyperpolarized membrane potential than that of the plasma membrane and other organelles and can be moved between cells [105,108]. During regeneration, the transfer of healthy mitochondria from glia (i.e., astrocytes) to neurons contributes to neuronal survival [109]. Likewise, in astrocytes, CO may have paracrine effects influencing mitochondrial biogenesis in astrocytes, leading to neuronal protection [110]. CO protects astrocytic apoptosis by inhibiting stimuli-mediated mitochondrial membrane permeabilization [111]. Astrocytic HO-1 overexpression increases mitochondrial biogenic gene expression, such as sirtuin 1 (SIRT1) and PGC-1α in a mouse model [15]. In CO/R conditioned astrocyte cells, Ca^2+^ entry via L-type Ca^2+^ channels can induce Ca^2+^-calmodulin kinase kinase β (CaMKKβ)-mediated AMPKα activation and these effects are initiated by CO and bilirubin [38]. CO/R stimulates the sequential AMPKα-SIRT1-PGC-1α axis in astrocytes, leading to enhanced ATP production and mitochondrial biogenesis [39]. Moreover, CO/R enhances circuits between HIF-1α and ERRα in astrocytes, leading to VEGF upregulation as well as mitochondrial biogenesis [40] (Figure 2).

The role of mitochondria in axon development and regeneration has emerged [105] and CO-mediated mitochondrial biogenesis may ascribe important roles in the mechanism of axon extension and regeneration as well as neurotransmitter releases. Thus, it is an aspect of CO’s CNS regeneration that needs further studies.

## 5. Signaling Pathways for Regeneration

In this section, important signaling pathways for regeneration in NSCs, neurons and endothelial cells will be introduced. CORMs may influence various signaling molecules, such as PI3K-Akt and MAPKs; those molecules contribute to CNS regeneration by stimulating angiogenesis, lymphangiogenesis and neurogenesis [112].

### 5.1. PI3K-Akt Pathway

CORM-2 upregulates PI3K-mediated Akt activation in astrocyte cells [37] and endothelial cells [113]. PI3K-Akt activation can induce endothelial cell survival and angiogenesis, as well as NSC survival and neurogenesis [114,115,116]. Akt phosphorylation by PI3Ks influences important diverse downstream signals, including endothelial NOS (eNOS), mTOR, glycogen synthase kinase-3β (GSK3β) and other signaling molecules [114,117] (Figure 2). In the CNS, genetic activation of the PI3K-Akt-mTOR pathway, through deficiencies in upstream negative regulators (i.e., phosphatase, PTEN, hamartin (tuberous sclerosis complex 1 (TSC1)) and tuberin (TSC2)), increases the axon regeneration capacity [118,119]. In PTEN-deleted RGC, overexpression of Armcx1 induces a substantial upregulation in the number of regenerating axons in comparison with PTEN deficiency alone [106].

Adhesion molecule with IgG-like domain 2 (AMIGO2), a novel membrane anchor of 3-phosphoinositide-dependent kinase 1 (PDK1), controls cell survival and angiogenesis via Akt activation [120]. In endothelial cells, loss of AMIGO2 led to apoptosis and inhibition of angiogenesis [120]. The AMIGO family with leucine-rich repeats is identified by ordered differential display, which reveals a novel sequence induced by amphoterin, the neurite-stimulating factor [121]. In addition, the role of AMIGO2 in neurovascular units has been demonstrated, such as neuronal development and neural circuit in zebrafish brain [122].

The angiopoietin-1-Tie2-PI3K axis initiates survival responses in neural progenitor cells after OGD [123]. PI3K activation can lead to protection of neurons and the brain from ischemic injury [124]. Further studies are necessary to describe how CO modulates the PI3K-Akt pathway and how their crosstalk contributes to CNS regeneration.

### 5.2. Mitogen-Activated Protein Kinase (MAPK)

The three main MAPK signaling pathways are the ERKs, p38 MAPK and c-Jun N-terminal kinases (JNKs) [125]. MEK-ERKs are considered important regulators of both angiogenesis and lymphangiogenesis [126]. Following early axon injury, ERK1 and ERK2 are phosphorylated and associated with both vimentin and locally translated importin-β in a Ca^2+^-dependent manner [127,128]. Those effects allow for binding to dynein for retrograde transport and inhibition from dephosphorylation, consequently regulating axon regeneration [6]. Following late axon injury, DLK enables the retrograde transport of several injury-related signaling proteins, including JNK, JNK-interacting protein 3 (JIP3) and DLK itself, to the nucleus [6,129,130]. Coordinated activation of JNK and p38 MAPK signaling is required for peripheral axon regeneration [131]. Collectively, the translocation of pro-regenerative transcription factors to the nucleus can initiate the expression of regenerative factors and consequent coordinated regulation of MAPK may promote regenerative pathways [6].

CORM-2 upregulates ERK1 and ERK2 phosphorylation in astrocytes, which are responsible for HIF-1α protein synthesis and the consequent upregulation of angiogenic VEGF secretion [37] (Figure 2). CO exposure upregulates p38 MAPK activation-mediated endothelial cells survival under inflammatory conditions and this effect is abolished by the inhibition of HO-1 [132]. Since phosphorylation-mediated activation of HIF-1α can be mediated by p38α, p38β and p38γ [60], CO may stabilize HIF-1α through the p38 MAPK pathway. Until now, the relationship between DLK-JNK signaling and CO has not been well established in axon regeneration.

## 6. Possible Regenerative Signaling Molecules through the Cell-Cell Network

Beyond the intrinsic pathway, other limitations of CNS repair appear to be the existence of inhibitory molecules, including semaphorin 3a, semaphorin 3f and netrin receptor Unc5b, that block the axon growth [133]. Those inhibitory molecules diminish the regenerative effects of neurotrophic factors secreted from glial cells, such as astrocytes, oligodendrocytes and microglia, on axon regeneration [134]. If the CO/HO-1 circuit affects various neurotrophic factors and suppresses inhibitory molecules concomitantly, it may boost neurovascular regeneration. In this section, we will discuss the role of CO in the release of neurotrophic factors and NO.

### 6.1. Ciliary Neurotrophic Factors (CNTFs)

Astrocytic expression of ciliary neurotrophic factor (CNTF) and leukemia inhibitory factor activates glycoprotein 130 (gp130, also known as IL6Rβ) in RGCs. In dorsal root ganglia neurons, IL-6 and CC-chemokine ligand 2 produced by macrophages are responsible for gp130 activation [135]. Activation of the JAK-STAT pathway by cytokines (i.e., CNTF) can increase RGC survival and moderately promote regeneration. The limited extent to which these cytokines can increase regeneration may be due to an injury-induced increase in a suppressor of cytokine signaling 3 (SOCS3) expression, a potent JAK-STAT inhibitor. Indeed, genetic deletion of SOCS3 significantly increases regeneration when combined with the expression of CNTF and knockout of the tumor suppressor PTEN. Simultaneous *Pten* and *Socs3* knockout greatly enhance ganglion neuronal intrinsic growth capacity [136].

Adeno-associated virus CNTF administration induces optic nerve regeneration, including RGC axon guidance and long-lasting neuroprotection in chronically injured mice [137]. Direct evidence showing the link between CO and CNTF expression has not been well studied in CNS. In the peripheral system, CNTFs activate the nuclear transcription factor NF-E2-related factor 2 (Nrf2) by phosphorylation at residue serine 40 through Akt activation under OGD following reoxygenation (OGD/R) conditions in myocardial cells [138]. Nrf2 is the transcription factor for HO-1 and Nrf2 silencing or knockout abolished CNTF-induced cytoprotection against OGD/R [138]. In rat hepatocytes, HO-1 expression can be upregulated by IL-6 via the JAK-STAT pathway [139]. In addition to the JAK-STAT pathway, IL-6 also regulates the activities of several members of the MAPK and PI3K-Akt cascades [140]. Thus, we suggest that further investigations are necessary for demonstrating the role of CO in CNTF-mediated CNS regeneration.

### 6.2. Brain-Derived Neurotrophic Factor (BDNF)

Glia-derived BDNF promotes neuronal survival, differentiation, synaptic plasticity and angiogenesis in pathophysiological brains [141]. Increased BDNF protein in astrocytes and microglia at the injury site is demonstrated following the spinal cord injury [142]. Secreted BDNF binds with its high-affinity receptor tropomyosin-related kinase B (TrkB) and upregulates JIP3 expression in the hippocampal neuron via CREB activation [143]. JIP3 acts as an adaptor molecule for the anterograde axonal transport of TrkB receptors trafficking to axon terminals [144]. Vector-mediated overexpression of TrkB administrated through intrapleural injection is effective for recovery following spinal cord injury [145].

Upregulation of BDNF in mouse retina plays an important role in neuroprotection; enhanced RGC survival after optic nerve crush is shown by in vivo imaging [146]. Induced adult hippocampal neurogenesis via injected P7C3 (pool 7, compound 3), a compound that enhances NPC survival and BDNF restore cognition in a 5xFAD (five AD-linked mutations expressing human amyloid protein precursor and presenilin-1 transgenes) AD mouse model [100]. HO metabolites can stimulate BDNF levels in the CNS, suggesting that HO-1-BDNF-TrkB signaling modulates neural circuits [112].

### 6.3. Glial Cell-Line Derived Neurotrophic Factor (GDNF)

GDNF mRNA levels can be detected in the developing brain and persist from the first postnatal day into adulthood [134,147]. The molecular complex GDNF and GDNF receptor family alpha 1 is critical for the morphological maturation of dendritic growth and synapse formation of postnatal and adult hippocampal neurons and is required for correct spatial pattern separation memory [147,148].

HO-1 induction by adenovirus containing the human *HO-1* gene in the substantia nigra of rats increases mRNA and protein levels of BDNF and GDNF [149]. HO-1-derived astrocytic GDNF secretion reduces neuronal cell death [150]. Lentiviral vectors expressing GDNF are delivered to the striatum of rats following Parkinson’s disease-like injury [151]. In the group of animals where GDNF expression is switched on during degeneration, neurons are rescued and there is a reversal of motor deficits; in contrast, in the group of animals where GDNF expression is switched on 21 d after the injury, neurons are not rescued [151]. Taken together, the time of GDNF upregulation by HO-1 during CNS diseases can be important for regeneration.

### 6.4. Nitric Oxide (NO)

NO and CO are endogenous gases produced by NOS and HO and they are freely diffusible through the cell membrane and possess neurogenic and angiogenic functions [10,152]. NO is produced by the reaction of L-arginine with NOS isoforms. Two constitutive isoforms (eNOS and neuronal NOS [nNOS]) via Ca^2+^ entry and an inducible NOS are enzymes that are expressed in a highly cell type-specific manner. Acting as an intercellular signal, nNOS-NO can trigger neurogenesis in mouse brain neural progenitor cells. BDNF upregulates nNOS protein levels, which can induce the maturation of neurons from neural progenitor cells [153].

CORM-2 can induce eNOS activation and consequent NO production through the PI3K-Akt pathway in endothelial cells [113]. CO-mediated NO production may be involved in the reduction of glial scar formation after TBI. Biochemically altered glial cells, called reactive glia, contribute to the formation of a glial scar near sites of injury, possibly acting as barriers to CNS axon regrowth [154]. Injection of CORM-3 in TBI mice exhibits reduced astrocytic scar formation detected by glial fibrillary acidic protein (GFAP) levels, which are recovered by a NOS inhibitor in vivo [9]. CORM-3-injected mice after TBI show significantly enhanced NSC proliferation, migration and neuronal differentiation, which effects are abolished by a NOS inhibitor in vivo [9]. Therefore, CO-mediated NOS activation demonstrates dual effects, which are the (1) reduction of reactive astrocytes and (2) induction of NSC proliferation and differentiation. Recent work has suggested that preventing astrocytic scar formation results in no axon regrowth and instead, aids CNS axon regeneration [133]. Collectively, the modulation of glial scar extent and transition of GFAP-positive cells into Nestin-positive NSCs by CO may be important for the heterogeneity of NSCs and the capacity of CNS regeneration.

## 7. Functional Recovery by CO/HO-1 Pathway in CNS Injury

The CNS can, following injury, spontaneously undergo adaptive reorganization that helps it regain function. Stimulation of regeneration in regions of injury may contribute to mature neural circuits, resulting in the restoration of brain functions. Low doses of CO may possess therapeutic potential by inducing transient HO-1 expression, modulating NOS activity and inhibiting NOX-mediated inflammatory responses [8,43,155]. During an injury, CO may induce various neurotrophic factors consequently leading to angiogenesis, neurogenesis and mitochondrial biogenesis in SVZ and SGZ (Figure 3A).

### 7.1. Stroke

Transfer of astrocytic-derived mitochondria into neurons may enhance neuroplasticity and behavioral improvement [109]. In recent studies, astrocytic HO-1 induction increases the protein levels of two transcription factors, such as HIF-1α and ERRα, in a mouse model of ischemia/reperfusion injury [38,39,40]. Treatment of human astrocyte cells with CO/R enhances HIF-1α stabilization by eliciting the sequential activation of the sequential Ca^2+^-AMPKα-PGC-1α-ERRα axis [40]. This pathway results in increases in mitochondrial biogenesis and oxygen consumption, resulting in transient low oxygen levels that stabilizes the HIF-1α protein [38,39]. Activation of astrocytic HIF-1α in the CNS may contribute to mitochondrial biogenesis, neuroprotection and angiogenesis after ischemic injury [96,156].

CORM-3 (4 mg/kg) is injected either before or 3 d after intracerebral hemorrhage and results in reduction of the levels of inflammatory factors, such as TNF-α [36]. In addition, CORM-3 injection 1 h after ischemia/reperfusion suppresses neuroinflammation and protects the blood-brain barrier [97]. CORM-3-mediated HO-1 activation protects rat astrocyte cells from inflammatory responses by suppressing IL-1β production [20]. HO-1 transgenic mice exhibit smaller infarct volumes, reduced tissue lipid peroxidation and increased levels of anti-apoptotic proteins, as compared with wild-type mice experiencing cerebral ischemia [157]. HO-1 is an upstream factor for HIF-1α stabilization in the periinfarct region of ischemic stroke brain of mice [40] and HIF-1α induction in ischemic neurons may play a therapeutic role by inducing VEGF, a potent angiogenic and neurogenic factor, in a stroke model [158,159]. Thus, the CO/HO circuit may activate the HIF-1α-VEGF axis and boost regeneration via mitochondrial biogenesis, angiogenesis and neurogenesis in ischemic injury. 

### 7.2. Traumatic Brain Injury (TBI)

TBI can lead to degenerative neurovascular damage, which results in vascular permeability, impaired memory and cognitive difficulties. A glutamate wave excites neurons via excess Ca^2+^ entry. Ca^2+^-dependent NOS activation and excess peroxynitrite production can lead to cytotoxicity in various CNS cells in the TBI brain [160]. An apoptotic marker is co-stained with pericyte marker in ipsilateral mice brains on day 3 after TBI, which is significantly reduced by CORM-3 or CO gas [9]. In addition, treatment of brain pericytes with CORM-3 suppress reactive oxygen species (ROS) production, consequently protecting neighboring cells from ROS-mediated cytotoxicity in a paracrine manner.

Moreover, CO can enhance neurogenesis via crosstalk between pericytes and NSCs in the late neuroplasticity phase of TBI. NSCs located in CORM-3-treated TBI mice brains, such as SVZ and DG, exhibited increases in the ability to synthesize, migrate and differentiate into mature neurons as compared with that in TBI mice brains [9] (Figure 3B,C). NSCs are near pericytes in the CORM-3-treated TBI brain and the enhancement of the in vitro differentiation of NSCs into mature neurons is demonstrated when NSCs are cultured with conditioned medium from CORM-3-treated pericytes under OGD [9]. Interestingly, those effects are blocked when cells treated with a NOS inhibitor in vitro and in vivo [9]. In conclusion, CO may provide a therapeutic approach for TBI by suppressing pericyte apoptosis, enhancing the signaling network with NSCs and facilitating NOS-mediated neurogenesis.

### 7.3. Multiple Sclerosis

Multiple sclerosis is known as an autoimmune disease, characterized by oligodendrocytes death by immune cells (i.e., T lymphocytes), leading to demyelination and loss of neural circuit [161]. To mimic the neuroinflammatory responses and demyelination in multiple sclerosis, experimental autoimmune encephalomyelitis (EAE) model is widely used [162]. Sensitization to myelin antigens of EAE in HO-1 deficient mice shows more demyelination and paralysis compared with wild-type mice [163]. HO-1 induction inhibits the proliferation of helper T cell, leading to suppression of myelin-reactive immune responses [163]. CORM-A1 (2 mg/kg) is injected for 30 consecutive days starting from the day of immunization to examine prolonged prophylactic role of CO in EAE model [164]. In that study, CORM-A1 improves the clinical features of EAE and also reduces infiltration of polymorphonucleated cells of spinal cord [164]. Moreover, single nucleotide polymorphisms in the genes of HO-1 and HO-2 may be slightly associated with increased risk for multiple sclerosis [165]. 

Neurotrophic factors such as CNTF may promote new myelin synthesis, consequently resulting in functional recovery in multiple sclerosis [166]. Injection of EAE mice with CNTF-overexpressing mesenchymal stem cells leads to reduced inflammation and facilitated myelination via oligodendrogenesis [166]. Further investigations are necessary to find out the molecular and cellular mechanisms how HO-CO pathway improves clinical outcome in EAE model and multiple sclerosis by demonstrating the role of CO in CNTF-mediated CNS regeneration.

### 7.4. Alzheimer’s Disease (AD)

AD is a chronic neurodegenerative disease partially related to mitophagy-mediated Aβ accumulation and tau pathology in neurons [167]. Overexpression of HO-1 in the AD brain may be malfunctional. HO-1 expression is detected in the neocortex and cerebral vessels in the human AD brain [168]. Recently, it has been found that in post-mortem brain tissues of AD subjects, the phosphorylation of HO-1 serine residues is locally increased in the hippocampus, compared to the levels observed in age-matched controls [169]. Consequently, HO-1 can become a target for oxidative posttranslational alterations of the protein structure resulting in functional impairment. In the mouse brain, long-term overexpression of HO-1 facilitates tau aggregation with iron-loading mediated phosphorylation of tau at serine 396 and serine 199/202 residues [170]. Cognitive abnormality with impaired neural circuits can be observed in long-term HO-1-overexpressing mice [19].

In contrast, transient HO-1 upregulation by CORMs may be beneficial for AD therapies because several in vitro experiments demonstrate cytoprotective effects of CO. Treatment of astrocytes with CORM-2 suppresses the NOX-derived ROS caused by Aβ_(1–42)_, which may be mediated by HO-1 induction [171]. CORM-2 can protect neurons from ROS-mediated apoptosis by inhibiting voltage-dependent K^+^ (Kv) channels (i.e., Kv_2.1_), which facilitates intracellular K^+^ efflux as an initiating step in the apoptotic cascade [172]. Aβ_(1–42)_-mediated neuronal cell toxicity can be diminished by transient HO-1 overexpression or CORM-2 [172]. Further, in vivo experiments showing the effects of CO on AD therapies will give insights into drug development.

## 8. Perspectives

Neurons in adult CNS injuries have only a limited capacity for regeneration, possibly due to metabolically disturbed intrinsic and extrinsic environments [2]. Nevertheless, neuron replacement by endogenous NSCs may be able to repair the injured tissues under favorable conditions supported by the functional vascular system. CO may have regenerative potential by regulating intrinsic and environmental signals through cell-cell interactions. Neurotrophic enrichment environment stimulated by CO may also contribute to neurovascular regeneration and functional neural circuits (Figure 3). Further studies revealing the regenerative effects of CO on CNS repair can help to elucidate the therapeutic mechanisms.

## Figures and Tables

**Figure 1 ijms-21-02273-f001:**
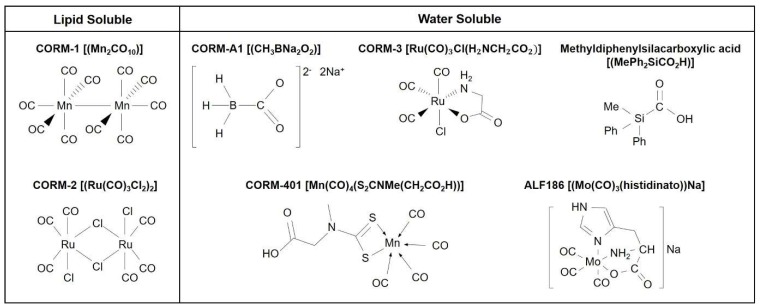
Structure and chemical components of CO-releasing molecules (CORMs).

**Figure 2 ijms-21-02273-f002:**
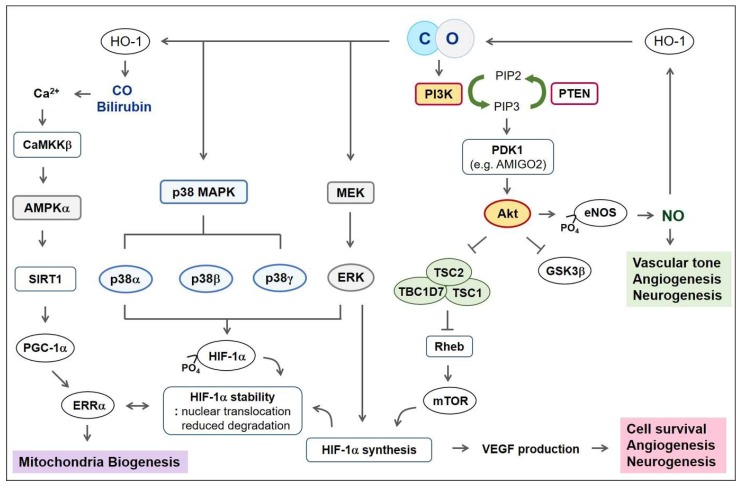
Schematic figures showing various signaling molecules influenced by the CO-HO-1 pathway. Firstly, CO can activate PI3K-Akt-eNOS-NO signaling. CO/NO crosstalk can lead to vessel dilation, angiogenesis and neurogenesis. NO-mediated HO-1 induction can produce CO, forming a positive feedback loop. Akt also inactivates GSK3β and the TBC1D7/TSC1/TSC2 complex. Then, activation of mTOR through inhibition of Rheb results in protein translation, such as HIF-1α. HIF-1α-mediated VEGF production and secretion induce cell survival, angiogenesis and neurogenesis. Secondly, through phosphorylation of HIF-1α by p38 MAPK and ERK, CO may stabilize HIF-1α resulting in its nuclear translocation and upregulation of ERRα expression. Thirdly, the combination of CO and bilirubin among HO-1 metabolites may initiate Ca^2+^ entry, consequently stimulating CaMKKβ-mediated AMPKα activation. AMPKα activates SIRT1, leading to PGC-1α deacetylation and de-ubiquitination. Stabilized PGC-1α upregulates ERRα, which are critical for mitochondrial biogenesis. Abbreviations: CO, carbon monoxide; HO, heme oxygenase; NO, nitric oxide; eNOS, endothelial NO synthase; PI3K, phosphatidylinositide 3-kinase; GSK3β, glycogen synthase kinase-3β; TSC1, hamartin; TSC2, tuberin; TBC1D7, tuberin Tre2-Bub2-Cdc16 domain family member 7; HIF-1α, Hypoxia-inducible factor-1α; MAPK, mitogen-activated protein kinase; ERK, extracellular-signal-related kinase; AMPKα, adenosine monophosphate kinase α; ERRα, estrogen-related receptor α; CaMKKβ, Ca^2+^-calmodulin kinase kinase β; SIRT1, sirtuin 1; PGC-1α, peroxisome proliferator-activated receptor γ-coactivator-1α.

**Figure 3 ijms-21-02273-f003:**
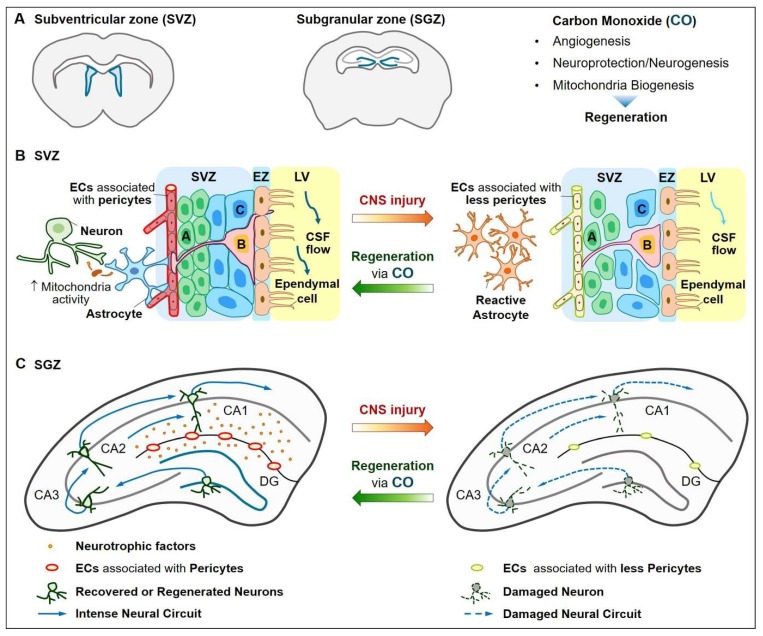
Schematic figures showing possible regenerative roles of CO in the adult neurovascular communications. (**A**) Adult neurogenesis occurs in the subventricular zone (SVZ) and subgranular zone (SGZ), possibly supported by functional vessels. CO can induce angiogenesis, neuroprotection/neurogenesis and mitochondrial biogenesis, consequently facilitating regeneration. (**B**) Among adult SVZ neural stem cells (NSCs), radial glia-like NSCs (B cells) reside along the ependymal zone and extend a radial process to contact blood vessels, which are associated with pericytes. B cells generate transit-amplifying cells (C cells). B cells also extend a cilium through the ependymal zone (EZ) to contact the cerebrospinal fluid (CSF) in the lateral ventricle (LV). Neuroblasts (A cells) migrate and differentiate into olfactory bulb neurons. In adult NSC niches, astrocytes contribute to the cellular architecture by connecting endothelial cells and neurons as well as transferring mitochondria. Functional communication can be disrupted by CNS injury and CO may have the capacity to recover cell-cell communications after injury. CO may reduce the extent of reactive astrocytes. (**C**) Intense neural circuit formation by CO may be possible after CNS injury through coordinated neurovascular networks. Endothelial cells associated pericytes may form the blood-brain barrier and stimulate the release of various growth/neurotrophic factors from surrounding glial cells, which are mediated by therapeutic doses of CO. In CO exposed conditions after CNS injury, newly synthesized neurons in the dentate gyrus (DG) may connect new neural circuits through the cornus ammonis (CA)3-CA2-CA1 pathway.

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
