# Peer review of "Regenerative Potential of Carbon Monoxide in Adult Neural Circuits of the Central Nervous System"

_ijms, 2020, doi:10.3390/ijms21072273_

Round 1

Reviewer 1 Report

The role of carbon monoxide as a signialling molecule is very much underappreciated in the literature. The review submitted by Jung et al. is timely and highly informative. The manuscript begins with an overview of heme oxygenase metabolite chemistry, which is followed by a detailed overview of the CO signalling pathways, which are then recapitulated in a context of therapeutic applications at the end. In summary, the manuscript is logically assembled, fairly comprehensive, and does an excellent job of highlighting the current gaps in knowledge that require attention.

General:

I have no issue with the content. The cited literature is very current.

The manuscript is very dense in abbreviations, therefore: 1) a list of abbreviations, ideally on the first page, would be extremely useful, and 2) to help with the text flow some terms such as ‘endothelial cells’, for example, should not be abbreviated even if they appear often.

It appears that there is an opportunity to add a section on multiple sclerosis in section 7 since it’s a major neurodegenerative disease and there is some work that has been done on HO-1 in the EAE model.

Specific:

Line 14-15 “CO may get directions…” This doesn’t make sense.

Line 14-15 “…the injured neurovascular system switches into regeneration mode by stimulating…”

Line 56 “…which can be found in tissues.” Which tissues? Perhaps give examples.

Lines 62-63 “…we will focus on the role of CO in neuronal regeneration” (or simply “in regeneration”

Line 169 The topic abruptly changes from the role of lymphatic vessels in waste removal to CNS neurogenesis. A sentence (or two) should be added to establish the link between topics.

Line 307 “…DLK enables the retrograde transport of several injury-related signalling proteins”

Line 337 Not certain is “coordinately” is used appropriately here.

Author Response

Response to Reviewer 1

Comment #1: The manuscript is very dense in abbreviations, therefore: 1) a list of abbreviations, ideally on the first page, would be extremely useful, and 2) to help with the text flow some terms such as ‘endothelial cells’, for example, should not be abbreviated even if they appear often.

Response #1: Thank you for your comments. (1) We added a list of abbreviations on the first and second page per your suggestion. (2) We changed ‘ECs’ into ‘endothelial cells’.

Comment #2: It appears that there is an opportunity to add a section on multiple sclerosis in section 7 since it’s a major neurodegenerative disease and there is some work that has been done on HO-1 in the EAE model.

Response #2: Thank you for your suggestion. We added a section on multiple sclerosis in section 7 (Line 543-560).

7.3 Multiple sclerosis

Multiple sclerosis is known as an autoimmune disease, characterized by oligodendrocytes death by immune cells (i.e., T lymphocytes), leading to demyelination and loss of neural circuit [1]. To mimic the neuroinflammatory responses and demyelination in multiple sclerosis, experimental autoimmune encephalomyelitis (EAE) model is widely used [2]. Sensitization to myelin antigens of EAE in HO-1 deficient mice shows more demyelination and paralysis compared with wild-type mice [3]. HO-1 induction inhibits the proliferation of helper T cell, leading to suppression of myelin-reactive immune responses [3]. CORM-A1 (2 mg/kg) is injected for 30 consecutive days starting from the day of immunization to examine prolonged prophylactic role of CO in EAE model [4]. In that study, CORM-A1 improves the clinical features of EAE and also reduces infiltration of polymorphonucleated cells of spinal cord [4]. Moreover, single nucleotide polymorphisms in the genes of HO-1 and HO-2 may be slightly associated with increased risk for multiple sclerosis [5].

Neurotrophic factors such as CNTF may promote new myelin synthesis, consequently resulting in functional recovery in multiple sclerosis [6]. Injection of EAE mice with CNTF-overexpressing mesenchymal stem cells leads to reduced inflammation and facilitated myelination via oligodendrogenesis [6]. Further investigations are necessary to find out the molecular and cellular mechanisms how HO-CO pathway improves clinical outcome in EAE model and multiple sclerosis by demonstrating the role of CO in CNTF-mediated CNS regeneration.

Specific:

Comment #3: Line 14-15 “CO may get directions…” This doesn’t make sense. / Line 14-15 “…the injured neurovascular system switches into regeneration mode by stimulating…”

Response #3: We changed that sentence. “CO may give directions by which the injured neurovascular system switches into regeneration mode by stimulating endogenous neural stem cells and endothelial cells to produce neurons and vessels capable of replacing injured neurons and vessels in the CNS.”

Comment #4: Line 56 “…which can be found in tissues.” Which tissues? Perhaps give examples.

Response #4: Line 126: we added “which can be found in nervous tissue [7].”

Comment #5: Lines 62-63 “…we will focus on the role of CO in neuronal regeneration” (or simply “in regeneration”

Response #5: Line 132-133: we changed that. “…. we will focus on the role of CO in regenerative potential.”

Comment #6: Line 169 The topic abruptly changes from the role of lymphatic vessels in waste removal to CNS neurogenesis. A sentence (or two) should be added to establish the link between topics.

Response #6: We added sentences for the link for topics. “Recently, the important role of lymphatic vessels in the drainage system for waste removal in the CNS has emerged [8-10]. Healthy lymphatic vessels remove macromolecules such as beta-amyloid (Ab), consequently maintaining functional CNS neural circuit [9, 11]. Aging and CNS injury diminish the clearance ability of lymphatic vessels [11-13].”

Comment #7: Line 307 “…DLK enables the retrograde transport of several injury-related signalling proteins”

Response #7: Per your suggestion, we changed that.

Line 377: “Following late axon injury, DLK enables the retrograde transport of several injury-related signaling proteins”

Comment #8: Line 337 Not certain is “coordinately” is used appropriately here.

Response #8: We removed “coordinately” in that sentence. Thank you for your comment.

References

  1. Fagone, P.; Patti, F.; Mangano, K.; Mammana, S.; Coco, M.; Touil-Boukoffa, C.; Chikovani, T.; Di Marco, R.; Nicoletti, F., Heme oxygenase-1 expression in peripheral blood mononuclear cells correlates with disease activity in multiple sclerosis. J Neuroimmunol 2013, 261, (1-2), 82-6.
  2. Picard-Riera, N.; Decker, L.; Delarasse, C.; Goude, K.; Nait-Oumesmar, B.; Liblau, R.; Pham-Dinh, D.; Baron-Van Evercooren, A., Experimental autoimmune encephalomyelitis mobilizes neural progenitors from the subventricular zone to undergo oligodendrogenesis in adult mice. Proc Natl Acad Sci U S A 2002, 99, (20), 13211-6.
  3. Chora, A. A.; Fontoura, P.; Cunha, A.; Pais, T. F.; Cardoso, S.; Ho, P. P.; Lee, L. Y.; Sobel, R. A.; Steinman, L.; Soares, M. P., Heme oxygenase-1 and carbon monoxide suppress autoimmune neuroinflammation. J Clin Invest 2007, 117, (2), 438-47.
  4. Fagone, P.; Mangano, K.; Quattrocchi, C.; Motterlini, R.; Di Marco, R.; Magro, G.; Penacho, N.; Romao, C. C.; Nicoletti, F., Prevention of clinical and histological signs of proteolipid protein (PLP)-induced experimental allergic encephalomyelitis (EAE) in mice by the water-soluble carbon monoxide-releasing molecule (CORM)-A1. Clin Exp Immunol 2011, 163, (3), 368-74.
  5. Agundez, J. A.; Garcia-Martin, E.; Martinez, C.; Benito-Leon, J.; Millan-Pascual, J.; Diaz-Sanchez, M.; Calleja, P.; Pisa, D.; Turpin-Fenoll, L.; Alonso-Navarro, H.; Pastor, P.; Ortega-Cubero, S.; Ayuso-Peralta, L.; Torrecillas, D.; Garcia-Albea, E.; Plaza-Nieto, J. F.; Jimenez-Jimenez, F. J., Heme Oxygenase-1 and 2 Common Genetic Variants and Risk for Multiple Sclerosis. Sci Rep 2016, 6, 20830.
  6. Lu, Z.; Hu, X.; Zhu, C.; Wang, D.; Zheng, X.; Liu, Q., Overexpression of CNTF in Mesenchymal Stem Cells reduces demyelination and induces clinical recovery in experimental autoimmune encephalomyelitis mice. J Neuroimmunol 2009, 206, (1-2), 58-69.
  7. Schipper, H. M., Heme oxygenase expression in human central nervous system disorders. Free radical biology & medicine 2004, 37, (12), 1995-2011.
  8. Louveau, A.; Smirnov, I.; Keyes, T. J.; Eccles, J. D.; Rouhani, S. J.; Peske, J. D.; Derecki, N. C.; Castle, D.; Mandell, J. W.; Lee, K. S.; Harris, T. H.; Kipnis, J., Structural and functional features of central nervous system lymphatic vessels. Nature 2015, 523, (7560), 337-41.
  9. Aspelund, A.; Antila, S.; Proulx, S. T.; Karlsen, T. V.; Karaman, S.; Detmar, M.; Wiig, H.; Alitalo, K., A dural lymphatic vascular system that drains brain interstitial fluid and macromolecules. J Exp Med 2015, 212, (7), 991-9.
  10. Ahn, J. H.; Cho, H.; Kim, J. H.; Kim, S. H.; Ham, J. S.; Park, I.; Suh, S. H.; Hong, S. P.; Song, J. H.; Hong, Y. K.; Jeong, Y.; Park, S. H.; Koh, G. Y., Meningeal lymphatic vessels at the skull base drain cerebrospinal fluid. Nature 2019, 572, (7767), 62-66.
  11. Da Mesquita, S.; Fu, Z.; Kipnis, J., The Meningeal Lymphatic System: A New Player in Neurophysiology. Neuron 2018, 100, (2), 375-388.
  12. Song, E.; Mao, T.; Dong, H.; Boisserand, L. S. B.; Antila, S.; Bosenberg, M.; Alitalo, K.; Thomas, J. L.; Iwasaki, A., VEGF-C-driven lymphatic drainage enables immunosurveillance of brain tumours. Nature 2020, 577, (7792), 689-694.
  13. Preston, J. E., Ageing choroid plexus-cerebrospinal fluid system. Microsc Res Tech 2001, 52, (1), 31-7.

Reviewer 2 Report

The review is well organized to discuss the CO role in neuroinflammatory  and neurodegenerative disease. Both signaling pathways and functions of CO/HO has been summarized. The schematic figures are clear to show how HO/CO contribute to mitochondrial biogenesis. I suggest the review published in current version.

Author Response

Thank you for your comment.

Reviewer 3 Report

 Regenerative potential of carbon monoxide in adult neural circuits of the central nervous system

The present review discusses the regenerative potential of CO in acute and chronic neuroinflammatory diseases of the CNS, such as stroke, traumatic brain injury, and Alzheimer’s disease, and the role of signaling pathways and neurotrophic factors.

This review is interesting, especially in connection with stroke, traumatic brain injury, and Alzheimer’s disease.

Nevertheless, this manuscript should be improve before publishing:

Generally

Please replace at the end of 7.2: 7.1; 6.3; 6.1; 4.2; “therefore” to another synonyms.

Please add a List of abbreviation to your manuscript.

Specially

 Keywords

Please add also to Keywords: stroke; Alzheimer’s disease; traumatic brain injury; CNS.

Line 27: please add references after: “ in the CNS of mammals“.

Line 29: please add references after: “ can be overcome “.

Line 31: please add references after: “ support axonal growth “.

Line 38: please add references after: “ of heme catabolism by HO “.

Line 40: please add references after: “ as electron 39 donors “.

Line 39: please write out:  NADH and NADPH.

Line 72: please write out:  ALF-186.

Line 78: please add references at the end of this sentence.

Line 115: please add references after: “ cAMP“.

Line 119: please write out: Ca2+-cAMP-PKA.  

Line 122: please write out: c-Jun.

Line 131: please write out: HIF-1α.

Line 135: please write out: HDAC5.

Line 135: please write out: Hsp90-HIF-1α.

Line 140: please write out: S727 and Y705.

Line 145: please add references at the end of this sentence.

Lines 165-167: please add references at the end of both sentences.

Line 172: please add references at the end of this sentence.

Line 177: please write out: VEGF.

Line 182: please write out: HIF-1α.

Line 190: please write out: PGC-1 α.

Lines 202-203: please write out: PGC-1 α and PGC-1α-Prox1-ERRα.

Line 210: please add references at the end of this sentence.

Line 220: please add references at the end of this sentence.

Line 225: please add references at the end of this sentence.

Line 234: please write out: TBI.

Lines 251-253: please write out: PI3K; PTEN; OGD.

Line 264: please write out: AMPKα-SIRT1-PGC-1α.

Line 266: please write out: HIF-1α-ERRα.

Line 282: please write out: mTOR.

Line 301: please add references at the end of this sentence.

Line 306: please write out: DLK.

Line 311: please add references at the end of this sentence.

Line 316: please write out: p38α/β/γ.

Figure 2 please include all abbreviations used there.  Please do it like this: Abbreviations:…

Lines 343-345: please write out: RGCs; DRG; gp130.

Line 353: please add references at the end of this sentence.

Lines 369-376:  please write out: TrkB; JIP3; P7C3; 5xFAD AD mouse model.

Line 386: please add references at the end of this sentence.

Line 426: please write out: ERRα.

Line 428: please write out: Ca2+-AMPKα-PGC-1α-ERRα.

Line 472: please write out: S396 and S199/202.

Line 477: please write out: Kv2.1.

Line 499: please write out: CA3-CA2-CA1.

Please replace at the end of 7.2: 7.1; 6.3; 6.1; 4.2; “therefore” to another synonyms.

Author Response

Response to Reviewer 3

Generally

Comment #1: Please replace at the end of 7.2: 7.1; 6.3; 6.1; 4.2; “therefore” to another synonyms.

Response #1: Thank you for your suggestion. We changed them.

4.2: Hence, CO may induce adult neuroprotection and neurogenesis via direct actions in NSCs and indirect actions through cell-cell interactions.

6.1: Thus, we suggest that further investigations are necessary for demonstrating the role of CO in CNTF-mediated CNS regeneration.

6.3: Taken together, the time of GDNF upregulation by HO-1 during CNS diseases can be important for regeneration.

7.1: Thus, the CO/HO circuit may activate the HIF-1a-VEGF axis and boost regeneration via mitochondrial biogenesis, angiogenesis, and neurogenesis in ischemic injury.

7.2: In conclusion, CO may provide a therapeutic approach for TBI by suppressing pericyte apoptosis, enhancing the signaling network with NSCs and facilitating NOS-mediated neurogenesis.

Comment #2: Please add a List of abbreviation to your manuscript.

Response #2: Thank you for your suggestion. Actually, another reviewer also comments that. We added a list of abbreviations on the first page.

Specially

Comment #3: Keywords

Please add also to Keywords: stroke; Alzheimer’s disease; traumatic brain injury; CNS.

Response #3: We added them per your suggestion. We also added “multiple sclerosis” to Keywords, because another reviewer give suggestion to add “multiple sclerosis” in section 7.

“Keywords: carbon monoxide; neurogenesis; angiogenesis; regeneration; stroke; traumatic brain injury; multiple sclerosis; Alzheimer’s disease; central nervous system”

Comment #4: Line 27: please add references after: “ in the CNS of mammals“. / Line 29: please add references after: “ can be overcome “. / Line 31: please add references after: “ support axonal growth “.

Response #4: Thank you for your comment. We added references per your suggestion. Line number has been changed because we added a list of abbreviation on Line 25-90.

Line 94-101: “In the peripheral nervous system and the central nervous system (CNS) of lower vertebrates, there is robust regeneration of severed axons [1-3]. However, regeneration following CNS injury in higher mammals is very poor [4]. Investigators have tried to discover precise mechanisms to augment the limited ability of CNS neurons to recover normal functions [5, 6]. The purpose of this work was to decipher the crucial barriers to regeneration, so that they may be overcome [7]. One critical question regarding the poor regeneration of central axons is whether the low recovery reflects an inability of neurons themselves to grow, or an inability of the environment to support axonal growth [5].”

Comment #5: Line 38: please add references after: “ of heme catabolism by HO “. / Line 40: please add references after: “ as electron 39 donors “. / Line 39: please write out:  NADH and NADPH.

Response #5: Thank you for your comment. We added references.

Line 107-109: “of heme catabolism by HO [8, 9]. Biliverdin is a tetrapyrrolic bile pigment that is further reduced to the antioxidant bilirubin by the enzyme biliverdin reductase using nicotinamide adenine dinucleotide (NADH) and NADH phosphate (NADPH) as electron donors [10, 11].”

We also added a list of abbreviations for NADPH on the first page.

Comment #6: Line 72: please write out:  ALF-186.

Response #6: I have searched for that; however, I could not find the full name of ALF-186. Instead, we added “CORM ALF-186” according to other papers [12, 13]. Please check Line 142.

Comment #7: Line 78: please add references at the end of this sentence.

Response #7: We added reference in Line 147. “This result is likely due to the distinct rates of CO release by these compounds, as well as the specific chemical reactivity of each CORM with cellular components [14].”

Comment #8: Line 115: please add references after: “ cAMP“.

Response #8: Thank you for your comment. We added references. Line 184: “(cAMP) [15]”

Comment #9: Line 119: please write out: Ca2+-cAMP-PKA. 

Response #9: Line 188-189: to show the consequent signaling cascade, we changed it into “sequential Ca2+-cAMP-PKA axis”.

Line 184 and Line186 show the full name of cAMP and PKA, respectively. We also added a list of abbreviations on the first and second page.

Comment #10: Line 122: please write out: c-Jun.

Response #10: Line 191: we thank you for your comment. We changed it into “transcription factor activator protein-1 (c-Jun)”.

Comment #11: Line 131: please write out: HIF-1α. / Line 135: please write out: HDAC5.

Response #11: Line 193, Line 199 and Line 200 demonstrate the full name of HDAC5, HIF-1α and Hsp90, respectively. We also added a list of abbreviations on the first and second page for them.

Comment #12: Line 135: please write out: Hsp90-HIF-1α.

Response #12: Line 205: we changed “Hsp90-HIF-1α” into “binding between Hsp90 and HIF-1a”.

Comment #13: Line 140: please write out: S727 and Y705.

Response #13: Line 211-212: We changed them. “JAKs phosphorylate tyrosine 705 residue and MEKs phosphorylates serine 727 residue”.

Comment #14: Line 145: please add references at the end of this sentence.

Response #14: Line 214: we have searched neuronal STAT3-HO-1 circuit; however, we could not find any. Thus, we could not add references. We therefore mentioned it “In neurons, however, the STAT3-HO-1 circuit has not been well studied.”

Comment #15: Lines 165-167: please add references at the end of both sentences. / Line 172: please add references at the end of this sentence.

Response #15: Line 234-236: we added references. “Some vessels and neurons in the adult brain and retina can be replaced by new ones in pathophysiologic conditions [16, 17]. New vessels are generated from existing vessels during the process of angiogenesis [18].”

Comment #16: Line 177: please write out: VEGF. / Line 182: please write out: HIF-1α. / Line 190: please write out: PGC-1α.

Response #16: Line 151, Line 199 and Line 254 demonstrate the full name of VEGF, HIF-1α, and PGC-1α, respectively. We also added a list of abbreviations on the first and second page for them.

Comment #17: Lines 202-203: please write out: PGC-1 α and PGC-1α-Prox1-ERRα.

Response #17: We added a list of abbreviations on the first and second page for them.

  1. Line 254 shows full name of PGC-1α; Line 255 shows full name of ERRα; Line 268 shows full name Prox1.
  2. Line 275-276: For clear expression, we changed “PGC-1α-Prox1-ERRα” into “protein complexes among PGC-1a, Prox1, and ERRa”.

Comment #18: Line 210: please add references at the end of this sentence. / Line 220: please add references at the end of this sentence. / Line 225: please add references at the end of this sentence.

Response #18: Thank you for your comments. We added them.

  1. Line 281-282: “Dysfunction of vascular and lymphatic systems with age can lead to the accumulation of toxic aggregates within the brain, consequently causing neurodegenerative diseases such as AD [19].”
  2. Line 291: “Besides the intrinsic pathway in axons, new neurons can be substituted into injured adult neurons through NSC proliferation and differentiation [20].”
  3. Line 298: “The neuroprotective role of CO in injury in vitro models has been reported [21-23].”

Comment #19:  Line 234: please write out: TBI.

Response #19: Line 128-129 demonstrate the full name of TBI. We also added a list of abbreviations on the first and second page.

Comment #20:  Lines 251-253: please write out: PI3K; PTEN; OGD

Response #20: Previous lines (Line 260, Line 322, and Line 308) demonstrate the full name of PI3K, PTEN, and OGD, respectively. We also added a list of abbreviations on the first and second page.

Comment #21:  Line 264: please write out: AMPKα-SIRT1-PGC-1α.

Response #21: Line 337: we changed “AMPKα-SIRT1-PGC-1α” into “sequential AMPKa-SIRT1-PGC-1a axis” for clear expression. That is signal transduction axis. We also previously mentioned full name of those proteins with abbreviated forms.

Comment #22: Line 266: please write out: HIF-1α-ERRα.

Response #22: Line 338-339: we changed it into “circuits between HIF-1a and ERRa” to show more clearly.

Comment #23: Line 282: please write out: mTOR.

Response #23: Line 323 shows full name of mTOR. We also added a list of abbreviations on the first and second page.

Comment #24: Line 301: please add references at the end of this sentence.

Response #24: Line 374-375: we added reference. “The three main MAPK signaling pathways are the ERKs, p38 MAPK, and c-Jun N-terminal kinases (JNKs) [24].”

Comment #25: Line 306: please write out: DLK.

Response #25: Line 185-186 demonstrate the full name of DLK. We also added a list of abbreviations on the first page for DLK.

Comment #26: Line 311: please add references at the end of this sentence.

Response #26: Line 384-385: we added reference. “consequent coordinated regulation of MAPK may promote regenerative pathways [6].”

Comment #27: Line 316: please write out: p38α/β/γ.

Response #27: Line 390: we changed “p38α/β/γ” into “p38a, p38b, and p38γ”.

Comment #28: Figure 2 please include all abbreviations used there.  Please do it like this: Abbreviations:…

Response #28: Thank you for your comment. We include all abbreviations in Figure 2 legend. Line 404-410: “Abbreviations: CO, carbon monoxide; HO, heme oxygenase; NO, nitric oxide; eNOS, endothelial NO synthase; PI3K, phosphatidylinositide 3-kinase; GSK3b, glycogen synthase kinase-3b; TSC1, hamartin; TSC2, tuberin; TBC1D7, tuberin Tre2-Bub2-Cdc16 domain family member 7; HIF-1a, Hypoxia-inducible factor-1a; MAPK, mitogen-activated protein kinase; ERK, extracellular-signal-related kinase; AMPKa, adenosine monophosphate kinase a; ERRa, estrogen-related receptor a; CaMKKb, Ca2+-calmodulin kinase kinase b; SIRT1, sirtuin 1; PGC-1a, peroxisome proliferator-activated receptor γ-coactivator-1a.” In addition, we added a list of abbreviations on the first and second page for them.

Comment #28: Lines 343-345: please write out: RGCs; DRG; gp130.

Response #28: Thank you for your comment. We added a list of abbreviations on the first and second page for them.

  1. Line 216 shows RGCs’ full name; Line 422 shows full name of gp130.
  2. We removed DRG in Line 423. Instead, we changed it into dorsal root ganglia, because it is mentioned one time in this manuscript.

Comment #29: Line 353: please add references at the end of this sentence.

Response #29: Line 433: We have searched direct evidence between CO and CNTF expression; however, we could not. Thus, we could not add references. We therefore mentioned it “Direct evidence showing the link between CO and CNTF expression has not been well studied in CNS.”

Comment #30: Lines 369-376:  please write out: TrkB; JIP3; P7C3; 5xFAD AD mouse model.

Response #30: Thank you for your comments.

  1. Full name of TrkB and JIP3 is demonstrated in Line 447 and Line 381. We added a list of abbreviations on the first and second page for TrkB and JIP3.
  2. Line 454: we changed P7C3 into “P7C3 (pool 7, compound 3)”.
  3. Line 455-456: we demonstrated 5xFAD as “5xFAD (five AD-linked mutations expressing human amyloid protein precursor and presenilin-1 transgenes)”.

Comment #31: Line 386: please add references at the end of this sentence.

Response #31: Line 464: we added reference. “Lentiviral vectors expressing GDNF are delivered to the striatum of rats following Parkinson’s disease-like injury [25].”

Comment #32: Line 426: please write out: ERRα.

Response #32: Line 506: ERRα has been mentioned in Line 255. We also added a list of abbreviations on the first page for ERRα.

Comment #33: Line 428: please write out: Ca2+-AMPKα-PGC-1α-ERRα.

Response #33: Line 508: we changed it into “sequential Ca2+-AMPKa-PGC-1a-ERRa axis” to show the signaling cascade from Ca2+ to ERRa.

Comment #34: Line 472: please write out: S396 and S199/202.

Response #34: Line 571: we changed them into “at serine 396 and serine 199/202 residues”

Comment #35: Line 477: please write out: Kv2.1.

Response #35: Line 576-577: we changed it into “inhibiting voltage-dependent K+ (Kv) channels (i.e., Kv2.1)”

Comment #36: Line 499: please write out: CA3-CA2-CA1.

Response #36: Line 598: we added full name of CA. “cornus ammonis (CA)3-CA2-CA1”.

Comment #37: Please replace at the end of 7.2: 7.1; 6.3; 6.1; 4.2; “therefore” to another synonyms.

Response #37: Thank you for your suggestion. We changed them.

4.2: Hence, CO may induce adult neuroprotection and neurogenesis via direct actions in NSCs and indirect actions through cell-cell interactions.

6.1: Thus, we suggest that further investigations are necessary for demonstrating the role of CO in CNTF-mediated CNS regeneration.

6.3: Taken together, the time of GDNF upregulation by HO-1 during CNS diseases can be important for regeneration.

7.1: Thus, the CO/HO circuit may activate the HIF-1a-VEGF axis and boost regeneration via mitochondrial biogenesis, angiogenesis, and neurogenesis in ischemic injury.

7.2: In conclusion, CO may provide a therapeutic approach for TBI by suppressing pericyte apoptosis, enhancing the signaling network with NSCs and facilitating NOS-mediated neurogenesis.

References

  1. Huebner, E. A.; Strittmatter, S. M., Axon regeneration in the peripheral and central nervous systems. Results Probl Cell Differ 2009, 48, 339-51.
  2. He, Z.; Jin, Y., Intrinsic Control of Axon Regeneration. Neuron 2016, 90, (3), 437-51.
  3. Chen, Z. L.; Yu, W. M.; Strickland, S., Peripheral regeneration. Annu Rev Neurosci 2007, 30, 209-33.
  4. Schwab, M. E.; Strittmatter, S. M., Nogo limits neural plasticity and recovery from injury. Curr Opin Neurobiol 2014, 27, 53-60.
  5. Tedeschi, A.; Bradke, F., Spatial and temporal arrangement of neuronal intrinsic and extrinsic mechanisms controlling axon regeneration. Curr Opin Neurobiol 2017, 42, 118-127.
  6. Mahar, M.; Cavalli, V., Intrinsic mechanisms of neuronal axon regeneration. Nature reviews. Neuroscience 2018, 19, (6), 323-337.
  7. Hilton, B. J.; Bradke, F., Can injured adult CNS axons regenerate by recapitulating development? Development 2017, 144, (19), 3417-3429.
  8. Kim, Y. M.; Pae, H. O.; Park, J. E.; Lee, Y. C.; Woo, J. M.; Kim, N. H.; Choi, Y. K.; Lee, B. S.; Kim, S. R.; Chung, H. T., Heme oxygenase in the regulation of vascular biology: from molecular mechanisms to therapeutic opportunities. Antioxid Redox Signal 2011, 14, (1), 137-67.
  9. Tenhunen, R.; Marver, H. S.; Schmid, R., The enzymatic conversion of heme to bilirubin by microsomal heme oxygenase. Proceedings of the National Academy of Sciences of the United States of America 1968, 61, (2), 748-55.
  10. Maines, M. D.; Trakshel, G. M., Purification and characterization of human biliverdin reductase. Arch Biochem Biophys 1993, 300, (1), 320-6.
  11. Tenhunen, R.; Marver, H. S.; Schmid, R., Microsomal heme oxygenase. Characterization of the enzyme. J Biol Chem 1969, 244, (23), 6388-94.
  12. Ulbrich, F.; Hagmann, C.; Buerkle, H.; Romao, C. C.; Schallner, N.; Goebel, U.; Biermann, J., The Carbon monoxide releasing molecule ALF-186 mediates anti-inflammatory and neuroprotective effects via the soluble guanylate cyclase ss1 in rats' retinal ganglion cells after ischemia and reperfusion injury. J Neuroinflammation 2017, 14, (1), 130.
  13. Stifter, J.; Ulbrich, F.; Goebel, U.; Bohringer, D.; Lagreze, W. A.; Biermann, J., Neuroprotection and neuroregeneration of retinal ganglion cells after intravitreal carbon monoxide release. PLoS One 2017, 12, (11), e0188444.
  14. Motterlini, R.; Foresti, R., Biological signaling by carbon monoxide and carbon monoxide-releasing molecules. Am J Physiol Cell Physiol 2017, 312, (3), C302-C313.
  15. Zaccolo, M.; Pozzan, T., CAMP and Ca2+ interplay: a matter of oscillation patterns. Trends Neurosci 2003, 26, (2), 53-5.
  16. Friedlander, M., Fibrosis and diseases of the eye. J Clin Invest 2007, 117, (3), 576-86.
  17. Ming, G. L.; Song, H., Adult neurogenesis in the mammalian brain: significant answers and significant questions. Neuron 2011, 70, (4), 687-702.
  18. Folkman, J., Angiogenesis. Annu Rev Med 2006, 57, 1-18.
  19. Da Mesquita, S.; Fu, Z.; Kipnis, J., The Meningeal Lymphatic System: A New Player in Neurophysiology. Neuron 2018, 100, (2), 375-388.
  20. Choi, Y. K.; Maki, T.; Mandeville, E. T.; Koh, S. H.; Hayakawa, K.; Arai, K.; Kim, Y. M.; Whalen, M. J.; Xing, C.; Wang, X.; Kim, K. W.; Lo, E. H., Dual effects of carbon monoxide on pericytes and neurogenesis in traumatic brain injury. Nat Med 2016, 22, (11), 1335-1341.
  21. Xie, Z.; Han, P.; Cui, Z.; Wang, B.; Zhong, Z.; Sun, Y.; Yang, G.; Sun, Q.; Bian, L., Pretreatment of Mouse Neural Stem Cells with Carbon Monoxide-Releasing Molecule-2 Interferes with NF-kappaB p65 Signaling and Suppresses Iron Overload-Induced Apoptosis. Cell Mol Neurobiol 2016, 36, (8), 1343-1351.
  22. Chen, K.; Gunter, K.; Maines, M. D., Neurons overexpressing heme oxygenase-1 resist oxidative stress-mediated cell death. Journal of neurochemistry 2000, 75, (1), 304-13.
  23. Scheiblich, H.; Bicker, G., Regulation of microglial migration, phagocytosis, and neurite outgrowth by HO-1/CO signaling. Dev Neurobiol 2015, 75, (8), 854-76.
  24. Johnson, G. L.; Lapadat, R., Mitogen-activated protein kinase pathways mediated by ERK, JNK, and p38 protein kinases. Science 2002, 298, (5600), 1911-2.
  25. Quintino, L.; Avallone, M.; Brannstrom, E.; Kavanagh, P.; Lockowandt, M.; Garcia Jareno, P.; Breger, L. S.; Lundberg, C., GDNF-mediated rescue of the nigrostriatal system depends on the degree of degeneration. Gene Ther 2019, 26, (1-2), 57-64.

Round 2

Reviewer 3 Report

This manuscript was very exacly improved according to my suggestions.